# Preparation of a ZnO Nanostructure as the Anode Material Using RF Magnetron Sputtering System

**DOI:** 10.3390/nano12020215

**Published:** 2022-01-10

**Authors:** Seokwon Lee, Yeon-Ho Joung, Yong-Kyu Yoon, Wonseok Choi

**Affiliations:** 1Department of Electrical Engineering, Hanbat National University, Daejeon 34158, Korea; dltjrdnjs000@naver.com; 2Department of Electronic Engineering, Hanbat National University, Daejeon 34158, Korea; yhjoung@hanbat.ac.kr; 3Department of Electrical & Computer Engineering, University of Florida, Gainesville, FL 32603, USA; ykyoon@ece.ufl.edu

**Keywords:** ZnO nanostructure, H_2_ reduction, lithium-ion battery, PECVD, RF-magnetron sputtering system

## Abstract

In this study, a four-inch zinc oxide (ZnO) nanostructure was synthesized using radio frequency (RF) magnetron sputtering to maximize the electrochemical performance of the anode material of a lithium-ion battery. All materials were grown on cleaned p-type silicon (100) wafers with a deposited copper layer inserted at the stage. The chamber of the RF magnetron sputtering system was injected with argon and oxygen gas for the growth of the ZnO films. A hydrogen (H_2_) reduction process was performed in a plasma enhanced chemical vapor deposition (PECVD) chamber to synthesize the ZnO nanostructure (ZnO NS) through modification of the surface structure of a ZnO film. Field emission scanning electron microscopy and atomic force microscopy were performed to confirm the surface and structural properties of the synthesized ZnO NS, and cyclic voltammetry was used to examine the electrochemical characteristics of the ZnO NS. Based on the Hall measurement, the ZnO NS subjected to H_2_ reduction had a higher electron mobility and lower resistivity than the ZnO film. The ZnO NS that was subjected to H_2_ reduction for 5 min and 10 min had average roughness of 3.117 nm and 3.418 nm, respectively.

## 1. Introduction

Many studies have recently been conducted on the viability of using transition metal oxides (Co, Ni, Cu, Mo, etc.) as anode materials for lithium-ion batteries. Transition metal oxides are attracting attention for such applications due to their specific capacity (which is two to three times higher than that of carbon) They are currently used as an anode material for conventional lithium-ion batteries [1,2,3,4,5,6,7,8,9,10,11,12,13,14,15]. Among the various transition metal oxides, zinc oxide (ZnO) is characterized by its low cost, easy manufacturing, chemical stability, large (60 mV) exciton binding energy, and 3.37 eV band gap [16,17]. When used as an anode material for lithium-ion batteries, ZnO has a high theoretical capacity of 978 mAhg^−1^ and a large lithium-ion diffusion coefficient [18,19,20,21]. However, given the low electrical conductivity of ZnO, as the charge/discharge cycle progresses, the large volume changes (228%) during the lithiation and delithiation processes reduce the stability of the specific capacity of ZnO [22,23].

The radio frequency (RF) magnetron sputtering system is an example of a physical vapor deposition method. When the RF power is supplied to the target, glow discharge occurs, which prompts the argon (Ar) atoms to conflict with the target. The target particles with smaller binding energy erupt due to the collision with Ar atoms. These erupted target particles adhere to the substrate and form a thin film [24,25].

In this paper, a ZnO film was deposited using an RF magnetron sputtering system, and ZnO nanostructures (ZnO NS) were prepared through hydrogen (H_2_) reduction. Among the various methods of compensating for the disadvantages of ZnO, one of the most effective methods is changing its structure. The reduction treatment was performed in an H_2_ gas atmosphere to widen the active area, and ZnO NS was used as the anode material.

## 2. Experiment Details

### 2.1. Preparation of Substrates and Copper (Cu) Layer Deposition

P-type silicon (Si, 100) was used to produce substrates. It was first cleaned to remove impurities. Then, the 2×2 cm Si substrates were added to a trichloroethylene (TCE) solvent and cleaned sequentially with ultrasonic cleaners (i.e., acetone, methanol, and deionized (DI) water) for 10 min each, after which they were dried using a nitrogen (N_2_) gas gun. The cleaned Si substrates were installed on the stage of the RF magnetron sputtering system and inserted into its chamber (I.T.S, Daejeon, Korea, PG600A-600W).

The base pressure of the sputtering chamber was $5\times 10^{-5}\;\mathrm{Torr}$, and the atmosphere of the chamber was composed of argon (Ar) gas. Pre-sputtering was performed for five minutes to remove impurities in the chamber. The Cu layer deposition process was carried out in the Ar atmosphere at a working pressure of $1.5\times 10^{-2}\;\mathrm{Torr}$ using a four-inch Cu target. The detailed deposition conditions for the Cu layer are shown in Table 1, and the schematic diagram of the RF sputtering system is shown in Figure 1.

### 2.2. ZnO Film Deposition

For the deposition of the ZnO film, after the deposition of the Cu layer of the Si substrate, the Si substrate was inserted into the chamber of the RF magnetron sputtering system. The base pressure of the chamber was $5\times 10^{-5}\;\mathrm{Torr}$ and the atmosphere of the chamber was made up of a mixture of Ar and oxygen (O_2_) gas at a ratio of 3:1. Pre-sputtering was performed for five minutes to remove impurities in the chamber, after which the ZnO film was deposited at 250 °C with 150 W RF plasma power using a four-inch ZnO target. The detailed conditions of the deposition of the ZnO film are shown in Table 2.

### 2.3. H_2_ Reduction Process

Plasma-enhanced chemical vapor deposition (PECVD, Woosin CryoVac, Uiwang, Korea, CVD-R2) was used for the H_2_ reduction of the ZnO film. The Cu-ZnO film previously prepared on the Si substrates was inserted into the stage of the PECVD chamber. The base pressure was set at $5\times 10^{-5}\;\mathrm{Torr}$, and the stage temperature at 600 °C. The ZnO NS was synthesized through H_2_ reduction of the ZnO film by injecting H_2_ into the PECVD chamber. The H_2_ injected into the chamber combined with the O_2_ in the ZnO film and caused a reduction reaction that escaped as water vapor. The H_2_ reduction process was performed for 5 min and 10 min for each sample. The PECVD schematic diagrams are shown in Figure 2, and the detailed H_2_ reduction processes are shown in Table 3.

### 2.4. Analysis

Field emission scanning electron microscopy (FE-SEM, Hitachi, Tokyo, Japan, S-4800), energy dispersive spectrometry (EDS, Oxford Instrument, Abingdon-on-Thames, United Kingdom, X-MAX), and atomic force microscopy (AFM, Park System, Suwon, Korea, XE-100) analysis were performed for the ZnO NS prepared on an Si wafer, and Hall measurement (ECOPIA, Anyang, Korea, HMS-3000) and cyclic voltammetry (CV, Autolab Instruments, Ionenstrasse, Switzerland, ECO CHEMIE PGSTAT 100) analysis were performed to determine the electrical conductivity and the oxidation-reduction characteristics. The CV test was performed the ZnO NS working electrode in 1M of sodium chloride (NaCl) electrolyte. The counter electrode was platinum (Pt) and the reference electrode was Ag/AgCl. The potential range of CV test was −0.8 V to 0 V, while the scan rate was 50 mV/s.

## 3. Results and Discussion

### 3.1. Field Emission Scanning Electron Microscopy (FESEM)

Figure 3 shows the surface and cross-section FESEM images of the ZnO NS. Figure 3a,b how the surface and cross-section, respectively, of the ZnO film deposited on the Cu layer before the H_2_ reduction process was performed. Figure 3c,d show the surface and cross-section, respectively, of the ZnO NS on which the H_2_ reduction process was performed for five minutes. During the H_2_ reduction process, the ZnO film was etched with H_2_ and its structure changed. Therefore, it can be confirmed that the surface area of H_2_ is wider than that of the ZnO film. In addition, Figure 3e,f show the surface and cross-section, respectively, of the ZnO nanostructure that was etched with H_2_ for 10 min.

### 3.2. Energy Dispersive Spectrometry (EDS)

The ratios of zinc (Zn) and O_2_ in the ZnO film and the ZnO NS were confirmed through EDS analysis. The EDS measurement area of the ZnO film to which the H_2_ reduction process was not applied is shown in Figure 4a. Figure 4b,c show the EDS measurement area of the ZnO NS surface on which the H_2_ reduction process was performed for 5 and 10 min, respectively. The weight ratio and the atomic ratio of Zn and O_2_, which were determined using EDS analysis, are shown in Table 4. The weight ratio and the atomic ratio of the ZnO film and the ZnO NS that changed during the H_2_ reduction process are shown in Figure 4d. The figure shows that the longer the H_2_ reduction process, the lower the proportions of O_2_ in the ZnO film and in the ZnO NS. This was caused by the reaction of H_2_ and O_2_ during the H_2_ reduction process, which escaped as water vapor.

### 3.3. Hall Measurement

Hall measurement was performed on the ZnO films and on the ZnO NS. The graph in Figure 5 shows the mobility and resistance of the ZnO film and the ZnO NS. It shows that the ZnO NS subjected to H_2_ reduction had higher electron mobility and lower resistivity than the ZnO film. Although ZnO NS that was subjected to H_2_ reduction for five minutes showed higher electron mobility than the ZnO film subjected to H_2_ reduction for 10 min, their resistivity values were similar. These results show that the low electrical conductivity of the ZnO film can be increased through H_2_ reduction.

### 3.4. Atomic Force Microscopy (AFM)

The surface roughness was confirmed through AFM analysis of the ZnO NS. The average roughness (Ra) of the ZnO NS that was subjected to H_2_ reduction for five minutes was 3.117 nm, and its root mean square roughness (Rq) was 3.970 nm. The ZnO NS that was subjected to H_2_ reduction for 10 min had an Ra of 3.418 nm and an Rq of 4.427 nm. The comparison of the surface roughness values showed that the surface of the ZnO NS that was subjected to H_2_ reduction for 10 min was rougher. Figure 6a,b show the three-dimensional (3D) AFM images of the ZnO NS on which H_2_ reduction was performed for five minutes and 10 min, respectively, and Table 5 summarizes the main parameters of the AFM analysis for the ZnO NS.

### 3.5. Cyclic Voltammetry (CV)

Figure 7 shows the CV curve of the ZnO NS. Figure 7a is the CV curve for the ZnO NS in which H_2_ reduction was performed for five minutes. One oxidation peak at −2.4 V and one reduction peak at −4.4 V were observed. On the other hand, on the CV curve of the ZnO NS on which H_2_ reduction was performed for 10 min, as shown in Figure 7b, one reduction peak was observed at −6.4 V but no oxidation peak was observed. On a CV curve, if only an oxidation peak or a reduction peak is observed, it indicates that an irreversible reaction is occurring rather than a reversible reaction. Accordingly, it can be confirmed that ZnO NS subjected to H_2_ reduction for 5 min exhibits a reversible reaction that can be utilized as an anode material for a lithium-ion battery. However, ZnO NS subjected to H_2_ reduction for 10 min is expected to be difficult to apply as an anode material for lithium-ion batteries.

## 4. Conclusions

In this study, a ZnO NS was prepared. When it was subjected to H_2_ reduction, its structure changed and it was confirmed to have had higher electrical conductivity than the ZnO film. In addition, its roughness value was measured and its CV curve was analyzed. The CV curve showed that its properties of oxidation-reduction reaction as the anode material improved when it was subjected to H_2_ reduction for five minutes. Therefore, the ZnO nanostructure can be applied as an anode material for lithium-ion batteries.

## Figures and Tables

**Figure 1 nanomaterials-12-00215-f001:**
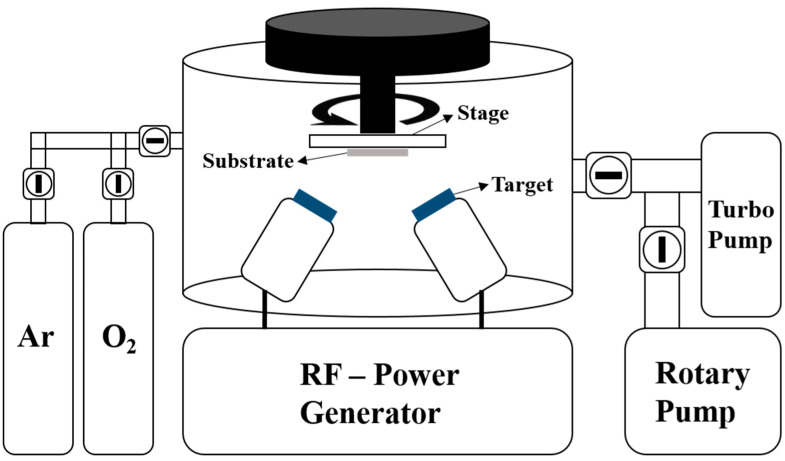
Schematic diagram of the RF magnetron sputtering system used in this study to synthesize Cu and ZnO.

**Figure 2 nanomaterials-12-00215-f002:**
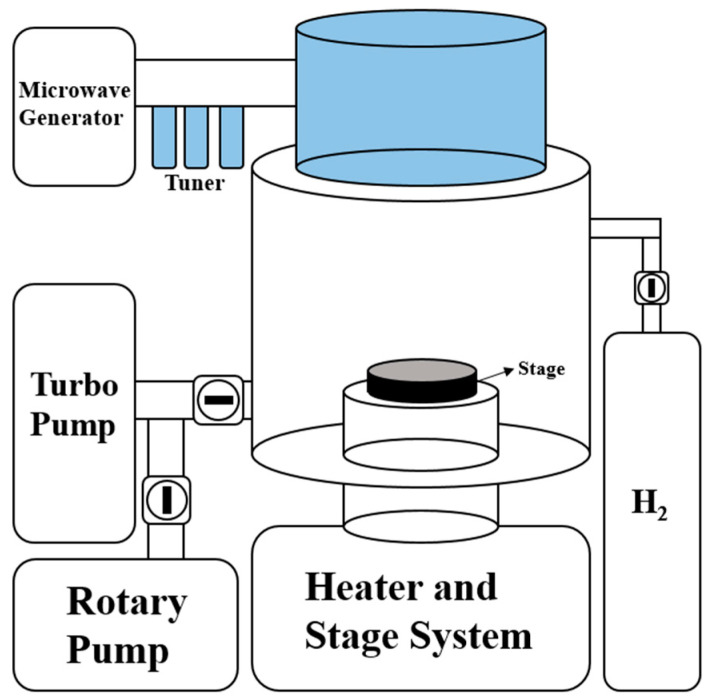
Schematic diagram of the PECVD used in this study to synthesize the ZnO NS using the H_2_ reduction process.

**Figure 3 nanomaterials-12-00215-f003:**
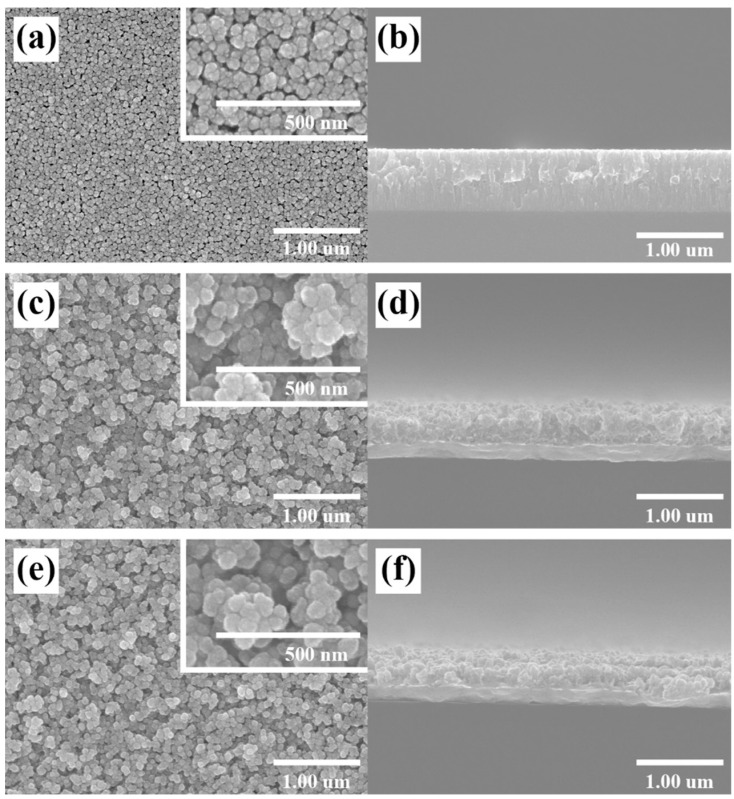
FESEM image of the surface (**a**) and cross-section (**b**) of the ZnO film; FESEM image of the surface (**c**) and cross-section (**d**) of the ZnO NS on which the H_2_ reduction process was performed for five minutes; FESEM image of the surface (**e**) and cross-section (**f**) of the ZnO NS on which the H_2_ reduction process was performed for 10 min.

**Figure 4 nanomaterials-12-00215-f004:**
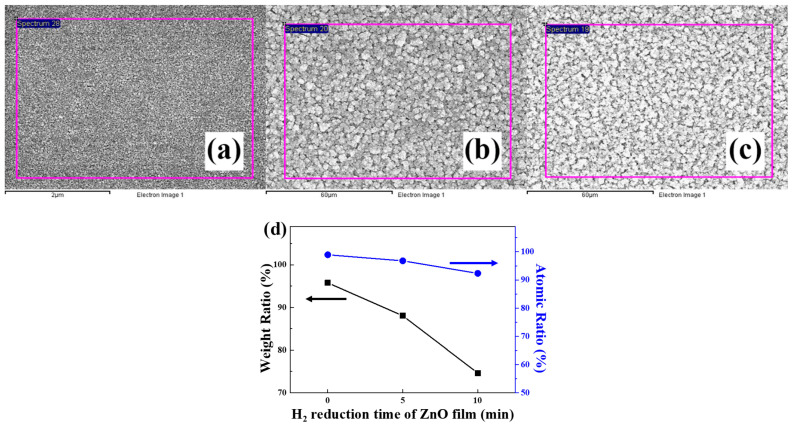
EDS analysis area of the ZnO film (**a**); EDS analysis area of the ZnO NS on which the H_2_ reduction process was carried out for five minutes (**b**) and 10 min (**c**); graph (**d**) of the ratios of Zn and O_2_ in the ZnO film and the ZnO NS.

**Figure 5 nanomaterials-12-00215-f005:**
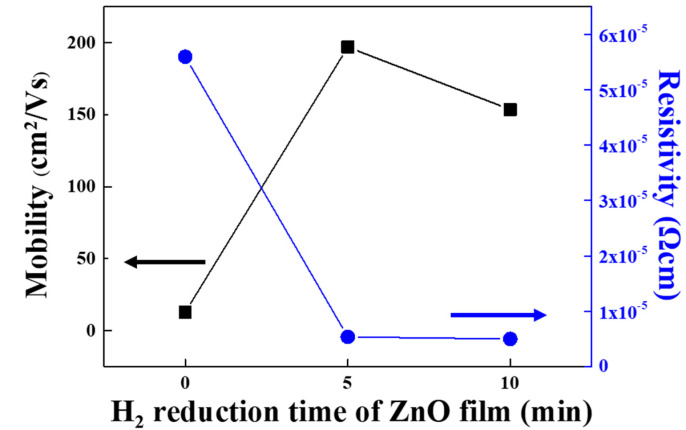
Hall measurement analysis to confirm the electrical conductivity of the ZnO film and the ZnO NS.

**Figure 6 nanomaterials-12-00215-f006:**
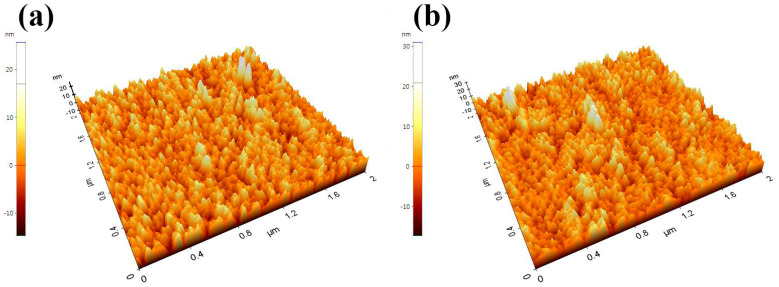
3D AFM analysis images of the ZnO NS on which the H_2_ reduction process was performed for five minutes (**a**) and 10 min (**b**).

**Figure 7 nanomaterials-12-00215-f007:**
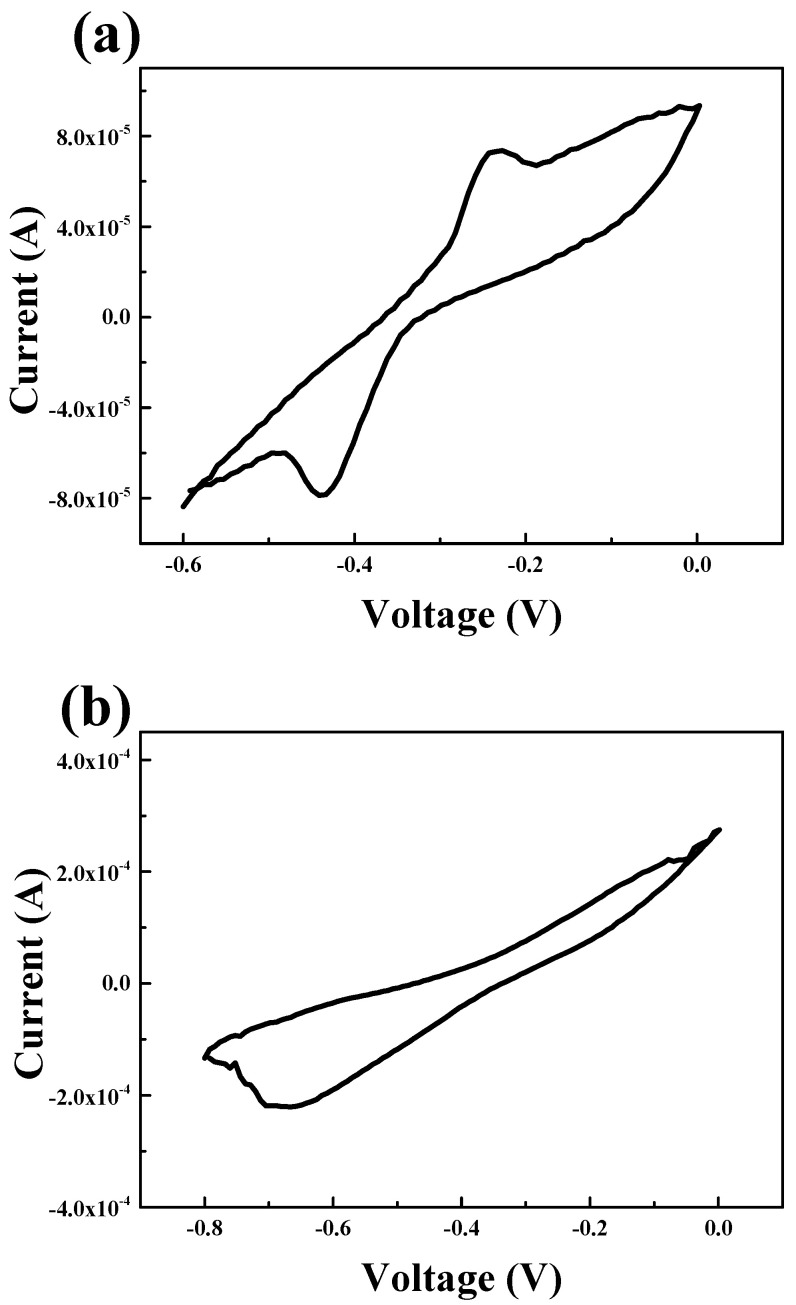
CV 2 of the ZnO NS on which the H_2_ reduction process was performed for 5 min (**a**) and 10 min (**b**).

**Table 1 nanomaterials-12-00215-t001:** Details of the Cu layer deposition conditions.

Substrate	Target	RF Power (W)	Base Pressure (Torr)	Working Pressure (Torr)	Deposition Time (min)	Temperature °C
Si (100)	Cu	300	$5\times 10^{-5}$	$1.5\times 10^{-2}$	15	250

**Table 2 nanomaterials-12-00215-t002:** Details of the ZnO film deposition conditions.

Substrate	Target	RF Power (W)	Base Pressure (Torr)	Working Pressure (Torr)	Deposition Time (min)	Temperature °C
Si wafer/Cu layer	ZnO	150	$5\times 10^{-5}$	$1.5\times 10^{-2}$	20	250

**Table 3 nanomaterials-12-00215-t003:** Details of the H_2_ reduction process conditions.

Samples	Substrate	H_2_ (sccm)	Base Pressure (Torr)	Temperature °C	Process Time (min)
Sample 1	Si wafer/Cu layer/ZnO film	100	$5\times 10^{-5}$	600	5
Sample 2	Si wafer/Cu layer/ZnO film	100	$5\times 10^{-5}$	600	10

**Table 4 nanomaterials-12-00215-t004:** Weight ratio and atomic ratio of Zn and O_2_ for the ZnO NS.

Samples	Zn Weight (%)	O_2_ Weight (%)	Zn Atomic (%)	O_2_ Atomic (%)
Cu-ZnO film	4.24	95.76	1.07	98.93
ZnO NS (5 min)	11.97	88.03	3.22	96.78
ZnO NS (10 min)	25.37	74.63	7.68	92.32

**Table 5 nanomaterials-12-00215-t005:** Parameters of the AFM analysis of the ZnO NS.

Samples	Ra	Rq	Skewness (Rsk)	Kurtosis (Rku)
ZnO NS (5 min)	3.117	3.970	−0.373	4
ZnO NS (10 min)	3.418	4.427	−0.612	4.721

## Data Availability

The data presented in this study are available in article.

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
