# Peer review of "Preparation of a ZnO Nanostructure as the Anode Material Using RF Magnetron Sputtering System"

_nanomaterials, 2022, doi:10.3390/nano12020215_

Round 1

Reviewer 1 Report

Lee et al studied the preparation of a ZnO nanostructure as the anode material using RF magnetron sputtering system. The work is interesting, but it cannot be considered for acceptance before making proper revisions. 

1. Radio frequency (RF) magnetron sputtering method was used in this work. So in the introduction part, the author should state the research progress of this technology and highlight the innovation of this paper.

2. The as-synthesized ZnO should be characterized in detail. XRD, XPS and other characterization toward Zno are recommended to be added into the revision.

3. Figure 7. The test conditions for CV curves should be described in detail in the experiment section.

4. Considering that ZnO is used as an anode in lithium-ion batteries, the lithium storage performance of the material should be tested in detail in this work. Moreover, It is suggested that the author add a table to compare the electrochemical data of this paper with that of other related papers.

5. Some references are too old, it is suggested to cite the latest achievements of ZnO in energy storage fields in recent three years. 
[1] CHEMICAL ENGINEERING JOURNAL, 2021, 425, 130660. DOI: 10.1016/j.cej.2021.130660.
[2] Journal of Power Sources, 2022, 518, 230761. doi:10.1016/j.jpowsour.2021.230761.
[3] NANOMATERIALS, 2021, 11, 2001. DOI: 10.3390/nano11082001.

Reviewer 2 Report

The manuscript entitled “Preparation of a ZnO Nanostructure as the Anode Material us-ing RF Magnetron Sputtering System”. The manuscript is well written, and the findings are innovative; however, it requires some refinement to improve the quality of the manuscript. Nonetheless, the manuscript can be further improved by considering the following aspects.

Main comments

  1. Abstract: Important information/parameters are missing, like the storage capacity, cycling characteristics and discharge capacity?
  2. Introduction: Please justify the novel work since ZnO structures are reported for anode materials. Please justify this point.
  3. The first paragraph contains trivial statements. The introduction should be reduced in length and focus on current analytical challenges. Essential related works can be cited.
  4. Introduction: It is better to include some recent references and application of ZnO structures: Journal of Power sources 488 (2021), 229393, Microchim Acta 186, 418 (2019), Energies 2021, 14(18), 5980, Appl. Sci. 2020, 10(17), 6062
  5. Section 2.2: what is the film thickness?
  6. It is better to provide XRD plot to confirm the ZnO crystalline behaviour and purity.
  7. The quality of some figures is inferior and needs to be enhanced.
  8. The quality of some figures is inferior and needs to be enhanced.
  9. It is better to check and correct the font size of the x and y-axis of all figures in the manuscript. It should be the same.
  10. It is better to include the Electrochemical performance comparison of ZnO-based anodes. Please include one comparison table.

Reviewer 3 Report

In this paper, a four-inch zinc oxide (ZnO) nanostructure was synthesized using radio frequency (RF) magnetron sputtering. The chamber of the RF magnetron sputtering system was injected with argon and oxygen gas for the growth of the ZnO films. Hydrogen (H2) reduction process was performed in plasma enhanced chemical vapor deposition (PECVD) chamber to synthesize ZnO nanostructure (ZnO NS) through modification of the surface structure of ZnO film. Therefore, the ZnO nanostructure can be applied as an anode material for lithium ion batteries. There are some issues which the authors should address them before acceptance process of the paper. Here are my comments:

  1. What are the advantages of this job over other jobs? The author is advised to make a table for comparison.
  2. The introduction is too simple. First, the author needs to supplement the existing preparation methods of ZnO films and analyze their advantages and disadvantages. Second, the author needs to supplement the application of ZnO film.
  3. Figure 1 the process of copper deposition should be a very mature process. The author does not need to draw a diagram alone.
  4. The scanning electron microscope shown in Fig. 3 does not match the atomic force scanning diagram shown in Fig. 6. The obvious scanning electron microscope shows that the particle roughness is very large, which is the stacking of zinc oxide particles, which needs to be explained by the author.
  5. I don't know why the zinc oxide treated by hydrogen can improve the electrochemical performance of lithium-ion battery. There is no connection between the two. The author needs to explain.
  6. For ZnO and its composite structure, relevant references need to be mentioned by the author, such as: Blue Luminescent ZnO Nanoclusters Stabilized by Esterifiable Polyamidoamine Dendrimers and their UV-Shielding; Preparation of ZnO/Ag2O Nanofibers by Coaxial Electrospinning and Study of Their Photocatalytic Properties; Preparation and Photoelectrocatalytic Performance of Fe2O3/ZnO Composite Electrode Loading on Conductive Glass; A near infrared fluorescence imprinted sensor based on zinc oxide nanorods for rapid determination of ketoprofen

Round 2

Reviewer 2 Report

All the comments are addressed properly, the manuscript is ready for the publication

Author Response

Thank you for your advice, which has improved the quality of our manuscript.

Reviewer 3 Report

The article has been systematically modified and can be accepted.

Author Response

(The authors gave the same response as above.)
